# Spectrum-Adapted Polynomial Approximation for Matrix Functions with Applications in Graph Signal Processing

**Tiffany  Fan [1]** , **David I Shuman [2],\*** , **Shashanka Ubaru [3]** and **Yousef Saad [4]**

[1]  Institute for Computational & Mathematical Engineering, Stanford University, Stanford, CA 94305, USA; tiffan@stanford.edu

[2]  Department of Mathematics, Statistics, and Computer Science, Macalester College, St. Paul, MN 55105, USA

[3]  IBM T.J. Watson Research Center, Yorktown Heights, NY 10598, USA; Shashanka.Ubaru@ibm.com

[4]  Department of Computer Science and Engineering, University of Minnesota, Minneapolis, MN 55455, USA; saad@umn.edu

\*  Correspondence: dshuman1@macalester.edu

**Abstract:**  We propose and investigate two new methods to approximate $f(\mathbf{A})\mathbf{b}$ for large, sparse, Hermitian matrices $\mathbf{A}$. Computations of this form play an important role in numerous signal processing and machine learning tasks. The main idea behind both methods is to first estimate the spectral density of $\mathbf{A}$, and then find polynomials of a fixed order that better approximate the function $f$ on areas of the spectrum with a higher density of eigenvalues. Compared to state-of-the-art methods such as the Lanczos method and truncated Chebyshev expansion, the proposed methods tend to provide more accurate approximations of $f(\mathbf{A})\mathbf{b}$ at lower polynomial orders, and for matrices $\mathbf{A}$ with a large number of distinct interior eigenvalues and a small spectral width. We also explore the application of these techniques to (i) fast estimation of the norms of localized graph spectral filter dictionary atoms, and (ii) fast filtering of time-vertex signals.

**Keywords:**  matrix function; spectral density estimation; polynomial approximation; orthogonal polynomials; graph spectral filtering; weighted least squares polynomial regression

## 1. Introduction

Efficiently computing $f(\mathbf{A})\mathbf{b}$, a function of a large, sparse Hermitian matrix times a vector, is an important component in numerous signal processing, machine learning, applied mathematics, and computer science tasks. Application examples include graph-based semi-supervised learning methods [1–3]; graph spectral filtering in graph signal processing [4]; convolutional neural networks/deep learning [5,6]; clustering [7,8]; approximating the spectral density of a large matrix [9]; estimating the numerical rank of a matrix [10,11]; approximating spectral sums such as the log-determinant of a matrix [12] or the trace of a matrix inverse for applications in physics, biology, information theory, and other disciplines [13]; solving semidefinite programs [14]; simulating random walks [15] (Chapter 8); and solving ordinary and partial differential equations [16–18].

References [19] (Chapter 13), [20–22] survey different approaches to this well-studied problem of efficiently computing

$$f(\mathbf{A})\mathbf{b} := \mathbf{V}f(\mathbf{\Lambda})\mathbf{V}^{\top}\mathbf{b}, \tag{1}$$

where the columns of $\mathbf{V}$ are the eigenvectors of the Hermitian matrix $\mathbf{A} \in \mathbb{R}^{N \times N}$; $\mathbf{\Lambda}$ is a diagonal matrix whose diagonal elements are the corresponding eigenvalues of $\mathbf{A}$, which we denote by

$\lambda_1, \lambda_2, \dots, \lambda_N$; and $f(\mathbf{\Lambda})$ is a diagonal matrix whose $k$th diagonal entry is given by $f(\lambda_k)$. For large matrices, it is not practical to explicitly compute the eigenvalues of $\mathbf{A}$ in order to approximate (1). Rather, the most common techniques, all of which avoid a full eigendecomposition of $\mathbf{A}$, include (i) truncated orthogonal polynomial expansions, including Chebyshev [23–25] and Jacobi; (ii) rational approximations [21] (Section 3.4); (iii) Krylov subspace methods such as the Lanczos method [23,26–29]; and (iv) quadrature/contour integral methods [19] (Section 13.3).

Our focus in this work is on polynomial approximation methods. Let $p_K(\lambda) = c_0 + \sum_{k=1}^{K} c_k \lambda^k$ be a degree $K$ polynomial approximation to the function $f$ on a known interval $[\underline{\lambda}, \overline{\lambda}]$ containing all of the eigenvalues of $\mathbf{A}$. Then the approximation $p_K(\mathbf{A})\mathbf{b}$ can be computed recursively, either through a three-term recurrence for specific types of polynomials (see Section 3 for more details), or through a nested multiplication iteration [30] (Section 9.2.4), letting $\mathbf{x}^{(0)} = c_K \mathbf{b}$, and then iterating

$$\mathbf{x}^{(l)} = c_{K-l}\mathbf{b} + \mathbf{A}\mathbf{x}^{(l-1)}, \; l = 1, 2, \dots, K. \tag{2}$$

The computational cost of either of these approaches is dominated by multiplying the sparse matrix $\mathbf{A}$ by $K$ different vectors. The approximation error is bounded by

$$||f(\mathbf{A}) - p_K(\mathbf{A})||_2 = \max_{\ell=1,2,\dots,N} |f(\lambda_\ell) - p_K(\lambda_\ell)| \tag{3}$$

$$\leq \sup_{\lambda \in [\underline{\lambda}, \overline{\lambda}]} |f(\lambda) - p_K(\lambda)|. \tag{4}$$

If, for example, $p_K$ is a degree $K$ truncated Chebyshev series approximation of an analytic function $f$, the upper bound in (4) converges geometrically to 0 as $K$ increases, at a rate of $\mathcal{O}\left(\rho^{-K}\right)$, where $\rho$ is the radius of an open Bernstein ellipse on which $f$ is analytic and bounded (see, e.g., [31] (Theorem 5.16), [32] (Theorem 8.2)).

In addition to the computational efficiency and convergence guarantees, a third advantage of polynomial approximation methods is that they can be implemented in a distributed setting [33]. A fourth advantage is that the $i$th element of $p_K(\mathbf{A})\mathbf{b}$ only depends on the elements of $\mathbf{b}$ within $K$ hops of $i$ on the graph associated with $\mathbf{A}$ (e.g., if $\mathbf{A}$ is a graph Laplacian matrix, a nonzero entry in the $(i, j)$th element of $\mathbf{A}$, where $i \neq j$, corresponds to an edge connecting vertices $i$ and $j$ in the graph). This localization property is important in many graph-based data analysis applications, such as graph spectral filtering [34] and deep learning [5]. Finally, as opposed to other methods that incorporate prior knowledge about $\mathbf{b}$ into the choice of the approximating polynomial (e.g., [35] considers vectors $\mathbf{b}$ from a zero-mean distribution with a known covariance matrix), the polynomial approximations resulting from the methods we propose do not depend on $\mathbf{b}$ or any information about $\mathbf{b}$. Thus, in applications where the computation of $f(\mathbf{A})\mathbf{b}$ is repeated for many different vectors $\mathbf{b}$ with the same $f$ and $\mathbf{A}$, the polynomial coefficients only need to be computed a single time.

While the classical truncated orthogonal polynomial expansion methods (e.g., Chebyshev, Legendre, Jacobi) aim to approximate the function $f$ throughout the full interval $[\underline{\lambda}, \overline{\lambda}]$, it is only the polynomial approximation error at the eigenvalues of $\mathbf{A}$ that affects the overall error in (3). With knowledge of the complete set of eigenvalues, we could do better, for example, by fitting a degree $K$ polynomial via the discrete least squares problem $\min_{p \in \mathcal{P}_K} \sum_{\ell=1}^{N} [f(\lambda_\ell) - p(\lambda_\ell)]^2$. In Figure 1, we show an example of such a discrete least squares fitting. The resulting approximation error $||f(\mathbf{A}) - p_K(\mathbf{A})||_2$ for $K = 5$ is 0.020, as opposed to 0.347 for the degree 5 truncated Chebyshev approximation. This is despite the fact that $\sup_{\lambda \in [\underline{\lambda}, \overline{\lambda}]} |f(\lambda) - p_K(\lambda)|$ is equal to 0.650 for the discrete least squares approximation, as opposed to 0.347 for the Chebyshev approximation.

While in our setting we do not have access to the complete set of eigenvalues, our approach in this work is to leverage recent developments in efficiently estimating the spectral density of the matrix $\mathbf{A}$, to adapt the polynomial to the spectrum in order to achieve better approximation accuracy at the (unknown) eigenvalues. After reviewing spectral density estimation in the next section, we present two new classes of spectrum-adapted approximation techniques in Section 3. In Section 4,

we perform numerical experiments, approximating $f(\mathbf{A})\mathbf{b}$ for different matrices $\mathbf{A}$ and functions $f$, and discuss the situations in which the proposed methods work better than the state-of-the-art methods. In Sections 5 and 6, we explore the application of the proposed technique to fast estimation of the norms of localized graph spectral filter dictionary atoms and fast filtering of time-vertex signals, respectively.

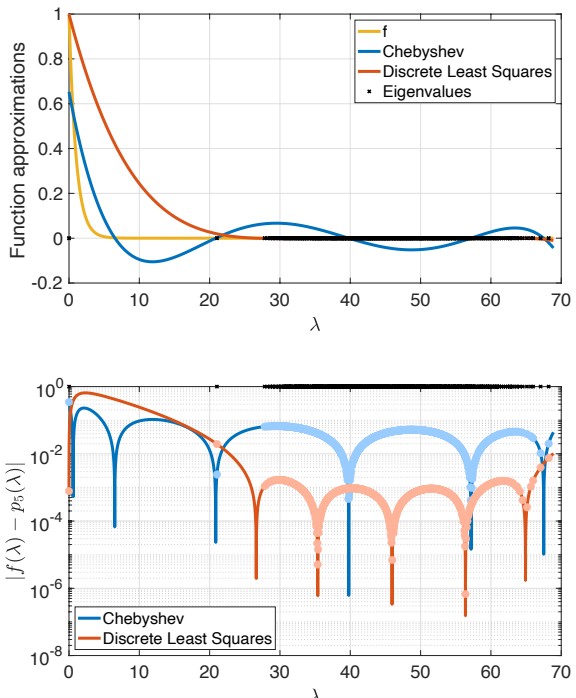

**Figure 1.** Degree 5 polynomial approximations of the function $f(\lambda) = e^{-\lambda}$ of the graph Laplacian of a random Erdös–Renyi graph with 500 vertices and edge probability 0.2. The discrete least squares approximation incurs larger errors in the lower end of the spectrum. However, since the eigenvalues are concentrated at the upper end of the spectrum, it yields a lower approximation error $||f(\mathbf{A}) - p_5(\mathbf{A})||_2$.

## 2. Spectral Density Estimation

The *cumulative spectral density function* or *empirical spectral cumulative distribution* of the matrix $\mathbf{A}$ is defined as

$$P_\lambda(z) := \frac{1}{N} \sum_{\ell=1}^{N} \mathbb{1}_{\{\lambda_\ell \leq z\}}, \tag{5}$$

where $\mathbb{1}_{\{C\}} = 1$ if statement $C$ is true and $\mathbb{1}_{\{C\}} = 0$ otherwise. The *spectral density function* [36] (Chapter 6) (also called the *Density of States* or *empirical spectral distribution* [37] (Chapter 2.4)) of $\mathbf{A}$ is the probability measure defined as

$$p_\lambda(z) := \frac{1}{N} \sum_{\ell=1}^{N} \mathbb{1}_{\{\lambda_\ell = z\}}.$$

Lin et al. [9] provide an overview of methods to approximate these functions. In this work, we use a variant of the Kernel Polynomial Method (KPM) [38–40] described in [9,41] to estimate the cumulative spectral density function $P_\lambda(z)$ of $\mathbf{A}$. Namely, for each of $S$ linearly spaced points $\{\xi_i\}_{i=1}^{S}$ between $\underline{\lambda}$ and $\overline{\lambda}$, we estimate the number of eigenvalues less than or equal to $\xi_i$ via stochastic trace estimation [42,43]. Let $\mathbf{x}$ denote a Gaussian random vector with distribution $\mathcal{N}(\mathbf{0}, \mathbf{I})$, $\{\mathbf{x}^{(j)}\}_{j=1}^{J}$

denote a sample of size $J$ from this distribution, and $\tilde{\Theta}_{\xi_i}$ denote a Jackson–Chebyshev polynomial approximation to $\Theta_{\xi_i}(\lambda) := \mathbb{1}_{\{\lambda \leq \xi_i\}}$ [44,45]. The stochastic trace estimate of the number of eigenvalues less than or equal to $\xi_i$ is then given by

$$\eta_i = \mathrm{tr}\big(\Theta_{\xi_i}(\mathbf{A})\big) = \mathbb{E}[\mathbf{x}^\top \Theta_{\xi_i}(\mathbf{A})\mathbf{x}] \approx \frac{1}{J}\sum_{j=1}^J \mathbf{x}^{(j)\top}\tilde{\Theta}_{\xi_i}(\mathbf{A})\mathbf{x}^{(j)}. \tag{6}$$

As in [46], we then form an approximation $\tilde{P}_\lambda(z)$ to $P_\lambda(z)$ by performing monotonic piecewise cubic interpolation [47] on the series of points $\left\{ \left(\xi_i, \frac{\eta_i}{N}\right) \right\}_{i=1,2,\dots,S}$. Analytically differentiating $\tilde{P}_\lambda(z)$ yields an approximation $\tilde{p}_\lambda(z)$ to the spectral density function $p_\lambda(z)$. Since $\tilde{P}_\lambda(z)$ is a monotonic cubic spline, we can also analytically compute its inverse function $\tilde{P}_\lambda^{-1}(y)$. The spectrum-adapted methods we propose in Section 3 utilize both $p_\lambda(z)$ and $\tilde{P}_\lambda^{-1}(y)$ to focus on regions of the spectrum with higher eigenvalue density when generating polynomial approximations. Figure 2 shows examples of the estimated cumulative spectral density functions for eight real, symmetric matrices $\mathbf{A}$: the graph Laplacians of the Erdös–Renyi graph (gnp) from Figure 1, the Minnesota traffic network [48] ($N = 2642$), and the Stanford bunny graph [49] ($N = 2503$); the normalized graph Laplacian of a random sensor network ($N = 5000$) from [50]; and the net25 ($N = 9520$), si2 ($N = 769$), cage9 ($N = 3534$), and saylr4 ($N = 3564$) matrices from the SuiteSparse Matrix Collection [51] (We use $\frac{\mathbf{A}+\mathbf{A}^\top}{2}$ for cage9, and for net25 and saylr4, we generate graph Laplacians based on the off-diagonal elements of $\mathbf{A}$. For saylr4, we scale the entire Laplacian by a factor of $\frac{1}{2000}$).

The computational complexity of forming the estimate $\tilde{P}_\lambda(z)$ is $\mathcal{O}(MJK_\Theta)$, where $M$ is the number of nonzero entries in $\mathbf{A}$, $J$ is the number of random vectors in (6) (in our experiments, $J = 10$ suffices), and $K_\Theta$ is the degree of the Jackson–Chebyshev polynomial approximations $\tilde{\Theta}_{\xi_i}$ [41]. While this cost is non-negligible if computing $f(\mathbf{A})\mathbf{b}$ for a single $f$ and a single $\mathbf{b}$, it only needs to be computed once for each $\mathbf{A}$ if repeating this calculation for multiple functions $f$ or multiple vectors $\mathbf{b}$, as is often the case in the applications mentioned above.

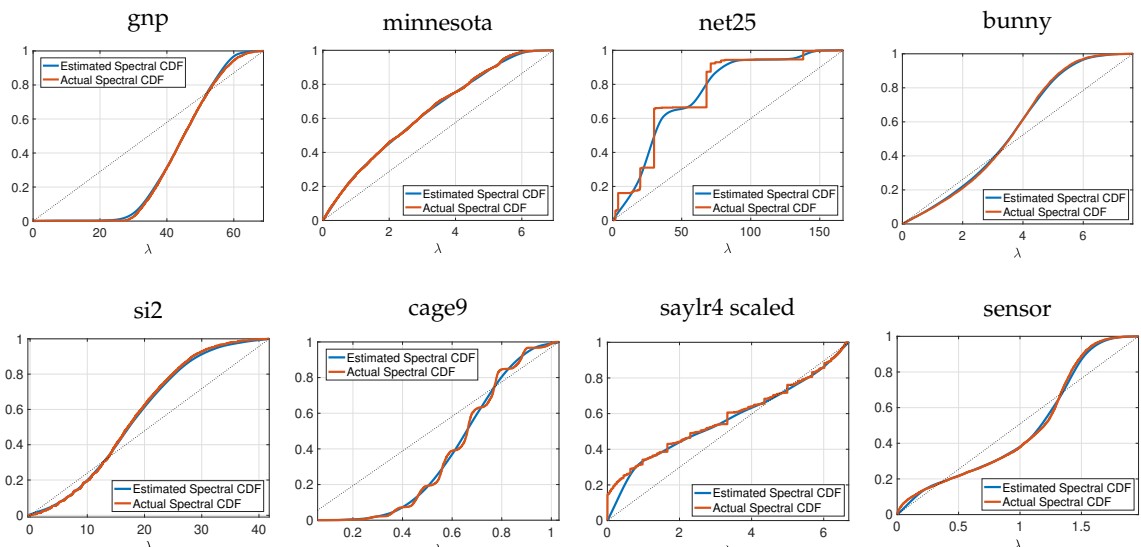

**Figure 2.** Estimated and actual cumulative spectral density functions for eight real, symmetric matrices $\mathbf{A}$. We estimate the eigenvalue counts for $S = 10$ linearly spaced points on $[\underline{\lambda}, \overline{\lambda}]$ via (6), with degree $K_\Theta = 30$ polynomials and $J = 10$ random vectors $\mathbf{x}^{(j)}$.

## 3. Spectrum-Adapted Methods

In this section, we introduce two new classes of degree $K$ polynomial approximations $p_K(\mathbf{A})\mathbf{b}$ to $f(\mathbf{A})\mathbf{b}$, both of which leverage the estimated cumulative spectral density function $\tilde{P}_\lambda(z)$.

### 3.1. Spectrum-Adapted Polynomial Interpolation

In the first method, we take $y_k := \frac{\cos(\frac{k\pi}{K})+1}{2}$, for $k = 0, 1, \ldots, K$, which are the $K+1$ extrema of the degree $K$ Chebyshev polynomial shifted to the interval $[0,1]$. We then warp these points via the inverse of the estimated cumulative spectral density function by setting $x_k = P_\lambda^{-1}(y_k)$, before finding the unique degree $K$ polynomial interpolation through the points $\{(x_k, f(x_k))\}_{k=0,1,\ldots,K}$. The intuition behind warping the interpolation points is that (i) a better approximation is attained in areas of the spectrum (domain) with more interpolation points, (ii) the error in (3) only depends on the errors at the eigenvalues of $A$, so we would like the approximation to be best in regions with many eigenvalues, and (iii) as shown in Figure 3, using the inverse of the estimated cumulative spectral density function as the warping function leads to a higher density of the warped points $\{x_k\}$ falling in higher density regions of the spectrum of $\mathbf{A}$. Thus, the warping should ideally lead to more interpolation points in high density regions of the spectrum, better approximation of the target function in these regions, and, in turn, a reduction in the error (3), as compared to interpolations generated from points spread more evenly across the spectrum.

To find the (unique) degree $K$ polynomial interpolation, our numerical implementation uses MATLAB's `polyfit` function, which centers and scales the data and then solves the resulting system of equations via a QR decomposition. Once the interpolating polynomial coefficients are attained, $p_K(\mathbf{A})\mathbf{b}$ can be computed, e.g., via (2) or by representing the interpolating polynomial as a linear combination of Chebyshev polynomials and using Chebyshev coefficients in the associated three-term recurrence [23,32]. This entire procedure is detailed in Algorithm 1.

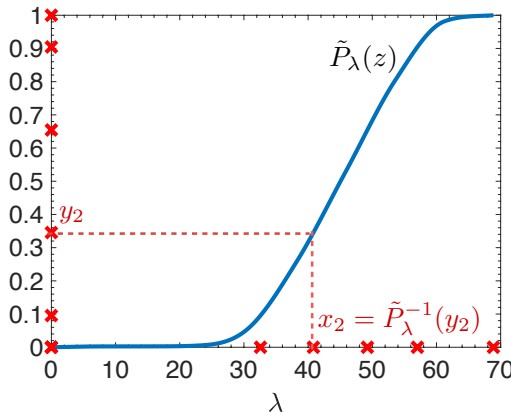

**Figure 3.** Construction of six interpolation points for the same graph Laplacian matrix described in Figure 1. The interpolation points $\{x_k\}$ on the horizontal axis are computed by applying the inverse of the estimated cumulative spectral density function to the initial Chebyshev points $\{y_k\}$ on the vertical axis.

---

**Algorithm 1** Spectrum-adapted polynomial interpolation.

---

**Input** Hermitian matrix $\mathbf{A} \in \mathbb{R}^{N \times N}$, vector $\mathbf{b} \in \mathbb{R}^N$, function $f$, polynomial degree $K$
**Output** degree $K$ approximation $p_K(\mathbf{A})\mathbf{b} \approx f(\mathbf{A})\mathbf{b} \in \mathbb{R}^N$

1: Compute $\tilde{P}_\lambda(z)$, an estimate of the cumulative spectral density of $\mathbf{A}$
2: **for** $k$ in $0 : K$ **do**
3:      $y_k \leftarrow \frac{1}{2}[\cos(\frac{k\pi}{K}) + 1]$
4:      $x_k \leftarrow \tilde{P}_\lambda^{-1}(y_k)$
5: **end for**
6: Find the (unique) degree $K$ polynomial interpolation $p_K$ through the points $\{(x_k, f(x_k))\}_{k=0,1,\ldots,K}$
7: Compute $p_K(\mathbf{A})\mathbf{b}$ via (2)
8: **return** $p_K(\mathbf{A})\mathbf{b}$

---

*3.2. Spectrum-Adapted Polynomial Regression/Orthogonal Polynomial Expansion*

A second approach is to solve the weighted least squares polynomial regression problem

$$\min_{p \in \mathcal{P}_K} \sum_{m=1}^{M} w_m \left[ f(x_m) - p(x_m) \right]^2, \tag{7}$$

where the abscissae $\{x_m\}_{m=1,2,\ldots,M}$ and weights $\{w_m\}_{m=1,2,\ldots,M}$ are chosen to capture the estimated spectral density function. We investigated several methods to set the points (e.g., linearly spaced points, Chebyshev points on the interval $[\underline{\lambda}, \overline{\lambda}]$, Chebyshev points on each subinterval $[\xi_i, \xi_{i+1}]$, and warped points via the inverse of the estimated cumulative spectral density function as in Section 3.1) and weights (e.g., the analytically computed estimate $\tilde{p}_\lambda$ of the spectral density function, a discrete estimate of the spectral density function based on the eigenvalue counts in (6), the original KPM density of states method based on a truncated Chebyshev expansion [9] (Equation 3.11), or equal weights for warped points). Without going into extensive detail about these various options, we remark that choosing abscissae $\{x_m\}$ that are very close to each other, which may occur when using points warped by the inverse of the estimated density function, may lead to numerical instabilities when solving the weighted least squares problem. In the numerical experiments, we use $M$ evenly spaced points on the interval $[\underline{\lambda}, \overline{\lambda}]$ (i.e., $x_m = \frac{m-1}{M-1}(\overline{\lambda} - \underline{\lambda}) + \underline{\lambda}$), and set the weights to be $w_m = \tilde{p}_\lambda(x_m)$. To solve (7), we use a customized variant of MATLAB's `polyfit` function that solves, again via QR decomposition, the normal equations of the weighted least squares problem:

$$\mathbf{\Psi}_\mathbf{x}^\top \mathbf{W} \mathbf{\Psi}_\mathbf{x} \mathbf{c} = \mathbf{\Psi}_\mathbf{x}^\top \mathbf{W} \mathbf{y},$$

where $\mathbf{\Psi}_\mathbf{x}$ is the Vandermonde matrix associated with the points $\{x_m\}$, $\mathbf{W}$ is a diagonal matrix with diagonal elements equal to the weights $\{w_m\}$, $\mathbf{y}$ is a column vector with entries equal to $\{f(x_m)\}$, and $\mathbf{c}$ is the vector of unknown polynomial coefficients. Once the coefficients $\mathbf{c}$ of the optimal polynomial, $p_K^*$ are attained, $p_K^*(\mathbf{A})\mathbf{b}$ can once again be computed, e.g., via (2). A summary of this method is detailed in Algorithm 2 (As pointed out by an anonymous reviewer, since our estimate $\tilde{p}_\lambda$ of the spectral density function is a piecewise quadratic function, the optimization problem (7) could be replaced by its continuous analog, $\min_{p \in \mathcal{P}_K} \int_{\underline{\lambda}}^{\overline{\lambda}} \left[ f(x) - p(x) \right]^2 \tilde{p}_\lambda(x) \, dx$, which can be solved analytically for many functions $f(\cdot)$).

---

**Algorithm 2** Spectrum-adapted polynomial regression.

---

**Input** Hermitian matrix $\mathbf{A} \in \mathbb{R}^{N \times N}$, vector $\mathbf{b} \in \mathbb{R}^N$, function $f$, polynomial degree $K$, number of grid points $M$

**Output** degree $K$ approximation $p_K(\mathbf{A})\mathbf{b} \approx f(\mathbf{A})\mathbf{b} \in \mathbb{R}^N$

1: Compute $\tilde{p}_\lambda(z)$, an estimate of the spectral density of $\mathbf{A}$
2: **for** $m$ in $1 : M$ **do**
3:     $x_m \leftarrow \frac{m-1}{M-1}(\overline{\lambda} - \underline{\lambda}) + \underline{\lambda}$
4:     $w_m \leftarrow \tilde{p}_\lambda(x_m)$
5: **end for**
6: $p_K \leftarrow \text{argmin}_{p \in \mathcal{P}_K} \sum_{m=1}^{M} w_m \left[ f(x_m) - p(x_m) \right]^2$
7: Compute $p_K(\mathbf{A})\mathbf{b}$ via (2)
8: **return** $p_K(\mathbf{A})\mathbf{b}$

---

An alternative way to view this weighted least squares method [52] is as a truncated expansion in polynomials orthogonal with respect to the discrete measure $d\lambda_M$ with finite support at the points $\{x_m\}$, and an associated inner product [53] (Section 1.1)

$$\langle f, g \rangle_{d\lambda_M} = \int_{\mathbb{R}} f(x)g(x)d\lambda_M = \sum_{m=1}^{M} w_m f(x_m)g(x_m).$$

The $M$ discrete monic orthogonal polynomials $\{\pi_{k,M}\}_{k=0,1,M-1}$ satisfy the three-term recurrence relation [53] (Section 1.3)

$$\pi_{k+1,M}(x) = (x - \alpha_{k,M})\pi_{k,M}(x) - \beta_{k,M}\pi_{k-1,M}(x), \ k = 0, 1, \ldots, M-1, \tag{8}$$

with $\pi_{-1,M}(x) = 0$, $\pi_{0,M}(x) = 1$, $\beta_{0,M} = \sum_{m=1}^{M} w_m$,

$$\alpha_{k,M} = \frac{\langle x\pi_{k,M}, \pi_{k,M} \rangle_{d\lambda_M}}{\langle \pi_{k,M}, \pi_{k,M} \rangle_{d\lambda_M}}, \ k = 0, 1, \ldots, M-1, \text{ and } \beta_{k,M} = \frac{\langle \pi_{k,M}, \pi_{k,M} \rangle_{d\lambda_M}}{\langle \pi_{k-1,M}, \pi_{k-1,M} \rangle_{d\lambda_M}}, \ k = 1, 2, \ldots, M-1.$$

Given the abscissae $\{x_m\}$ and weights $\{w_m\}$, the three-term recursion coefficients $\{\alpha_{k,M}\}_{k=0,1,\ldots,M-1}$ and $\{\beta_{k,M}\}_{k=1,2,\ldots,M-1}$ can also be computed through a stable Lanczos type algorithm on an $(M+1) \times (M+1)$ matrix [53] (Section 2.2.3), [54]. In matrix-vector notation, the vectors $\pi_{k,M} \in \mathbb{R}^M$, which are the discrete orthogonal polynomials evaluated at the $M$ abscissae, can be computed iteratively by the relation

$$\pi_{k+1,M} = (\text{diag}(\{x_m\}) - \alpha_{k,M}\mathbf{I}_M)\pi_{k,M} - \beta_{k,M}\pi_{k-1,M}, \ k = 0, 1, \ldots, M-1,$$

with $\pi_{-1,M} = \mathbf{0}_M$ and $\pi_{0,M} = \mathbf{1}_M$. Figure 4 shows an example of these discrete orthogonal polynomials.

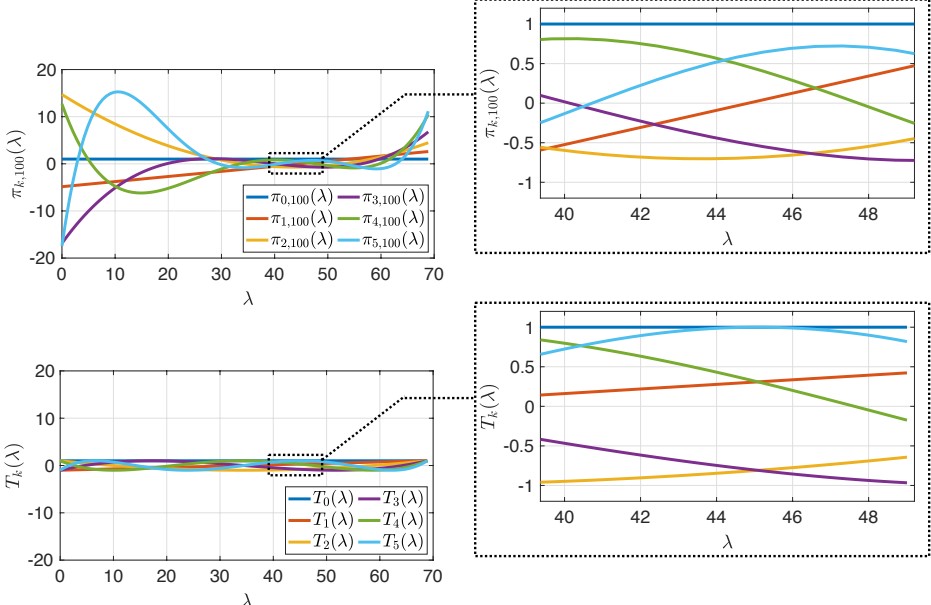

**Figure 4.** Comparison of (top) the first six discrete orthogonal polynomials defined in (8), adapted to the estimated cumulative spectral density of the random Erdös–Renyi graph from Figures 1–3, to (bottom) the first six standard shifted Chebyshev polynomials with degree $k = 0$ to 5. In the region of high spectral density, shown in the zoomed boxes on the right, the discrete orthogonal polynomials feature more oscillation while maintaining small amplitudes, enabling better approximation of smooth functions in this region.

Finally, the degree $K$ polynomial approximation to $f(\mathbf{A})\mathbf{b}$ is computed as

$$p_K(\mathbf{A})\mathbf{b} = \sum_{k=0}^{K} \frac{\langle f, \pi_{k,M} \rangle_{d\lambda_M}}{\langle \pi_{k,M}, \pi_{k,M} \rangle_{d\lambda_M}} \pi_{k,M}(\mathbf{A})\mathbf{b},$$

with $\pi_{-1,M}(\mathbf{A})\mathbf{b} = \mathbf{0}_N$, $\pi_{0,M}(\mathbf{A})\mathbf{b} = \mathbf{b}$, and

$$\pi_{k+1,M}(\mathbf{A})\mathbf{b} = (\mathbf{A} - \alpha_{k,M}\mathbf{I}_N)\pi_{k,M}(\mathbf{A})\mathbf{b} - \beta_{k,M}\pi_{k-1,M}(\mathbf{A})\mathbf{b}, \; k = 0, 1, \ldots, K-1 \text{ (where } K \leq M-1).$$

Before proceeding to numerical experiments, we briefly comment on the relationship between the spectrum-adapted approximation proposed in this section and the Lanczos approximation to $f(\mathbf{A})\mathbf{b}$, which is given by [19] (Section 13.2), [23]

$$\mathbf{Q}_K f(\mathbf{T}_K)\mathbf{Q}_K^\top \mathbf{b} = ||\mathbf{b}||_2 \mathbf{Q}_K f(\mathbf{T}_K)\mathbf{e}_1, \tag{9}$$

where $\mathbf{Q}_K$ is an $N \times (K+1)$ matrix whose columns form an orthonormal basis for $\mathcal{K}_K(\mathbf{A}, \mathbf{b}) = \text{span}\{\mathbf{b}, \mathbf{Ab}, \ldots, \mathbf{A}^K\mathbf{b}\}$, a Krylov subspace. In (9), $\mathbf{T}_K = \mathbf{Q}_K^\top \mathbf{A} \mathbf{Q}_K$ is a $(K+1) \times (K+1)$ tridiagonal Jacobi matrix. The first column of $\mathbf{Q}_K$ is equal to $\frac{\mathbf{b}}{||\mathbf{b}||}$. The approximation (9) can also be written as $q_K(\mathbf{A})\mathbf{b}$, where $q_K$ is the degree $K$ polynomial that interpolates the function $f$ at the $K+1$ eigenvalues of $\mathbf{T}_K$ [19] (Theorem 13.5), [55]. Thus, unlike classical polynomial approximation methods, such as the truncated Cheybshev expansion, the Lanczos method is indirectly adapted to the spectrum of $\mathbf{A}$. The Lanczos method differs from proposed method in that $\mathbf{T}_K$ and the Lanczos approximating polynomial $q_K$ depend on the initial vector $\mathbf{b}$. Specifically, the polynomials $\{\tilde{\pi}_k\}$ generated from the three-term recurrence of the form (8)

$$\gamma_{k+1}\tilde{\pi}_{k+1}(x) = (x - \alpha_k)\tilde{\pi}_k(x) - \gamma_k\tilde{\pi}_{k-1}(x),$$

with the $\{\alpha_k\}_{k=0,1,\ldots,K}$ and $\{\gamma_k\}_{k=1,2,\ldots,K}$ coefficients taken from the diagonal and superdiagonal entries of $\mathbf{T}_K$, respectively, are orthogonal with respect to the piecewise-constant measure

$$\mu(x) = \begin{cases} 0, & x < \lambda_1 \\ \sum_{j=1}^{i}[\hat{\mathbf{b}}(j)]^2, & \lambda_i \leq x < \lambda_{i+1} \\ \sum_{j=1}^{N}[\hat{\mathbf{b}}(j)]^2 = 1, & \lambda_N \leq x \end{cases},$$

where $\hat{\mathbf{b}} = \mathbf{V}^\top \mathbf{q}_1 = \mathbf{V}^\top \left( \frac{\mathbf{b}}{||\mathbf{b}||} \right)$, and $\hat{\mathbf{b}}(j)$ is its $j$th component [56] (Theorem 4.2). If $\hat{\mathbf{b}}$ happens to be a constant vector, then $\mu(x) = P_\lambda(x)$ from (5). If $\mathbf{A}$ is a graph Laplacian, $\hat{\mathbf{b}}$ is the graph Fourier transform [4] of $\mathbf{b}$, normalized to have unit energy.

## 4. Numerical Examples and Discussion

In Figure 5, for different functions $f(\lambda)$ and matrices $\mathbf{A}$, we approximate $f(\mathbf{A})\mathbf{b}$ with $\mathbf{b} = \mathbf{V1}$ and polynomial approximation orders ranging from $K = 3$ to 25. To estimate the cumulative spectral density function $\tilde{P}_\lambda(z)$ with parameters $S = 10$, $J = 10$, and $K_\Theta = 30$, we use the KPM, as shown in Figure 2. Based on the analytical derivative and inverse function of $\tilde{P}_\lambda(z)$, we obtain the two proposed spectrum-adapted polynomial approximations for $f(\lambda)$, before computing each $p_K(\mathbf{A})\mathbf{b}$ via the corresponding three-term recursion. We compare the proposed methods to the truncated Chebyshev expansion and the Lanczos method with the same polynomial order. Note that when $\mathbf{b}$ is a constant vector in the spectral domain of $\mathbf{A}$, the relative error $\frac{||f(\mathbf{A})\mathbf{b} - p_K(\mathbf{A})\mathbf{b}||_2^2}{||f(\mathbf{A})\mathbf{b}||_2^2}$ is equal to $\frac{\sum_{\ell=1}^{N}(f(\lambda_\ell) - p_K(\lambda_\ell))^2}{\sum_{\ell=1}^{N}f(\lambda_\ell)^2}$, the numerator of which is the discrete least squares objective mentioned in Section 1. The first column of Figure 5 displays the errors at all eigenvalues of $\mathbf{A}$ for each order 10 polynomial

approximation of $f(\lambda) = e^{-\lambda}$. The second column examines the convergence of relative errors in approximating $e^{-\mathbf{A}}\mathbf{b}$ for matrices with various spectral distributions, for each of the four methods.

In the field of graph signal processing [4], it is common to analyze or modify a *graph signal* $\mathbf{b} \in \mathbb{R}^N$, where $b(i)$ is the value of the graph signal at vertex $i$ of a weighted, connected graph $G$ with $N$ vertices, by applying a graph spectral filter $f_l$. The filtered signal is exactly the product (1) of a function of the graph Laplacian (or some other symmetric matrix) and a vector, the graph signal $\mathbf{b}$. In Figure 6, we show examples of collections of such functions, commonly referred to as *graph spectral filter banks*. In the right three columns of Figure 5, we examine the relative errors incurred by approximating $f_l(\mathbf{A})\mathbf{b}$ for the lowpass, bandpass, and highpass graph spectral filters $f_1$, $f_3$, and $f_5$ shown in Figure 6a. We expand on the applications of such graph spectral filters in Section 5.

We make two observations based on the numerical examples:

1. The spectrum-adapted interpolation method often works well for low degree approximations ($K \leq 10$), but is not very stable at higher orders due to overfitting of the polynomial interpolant to the specific $K+1$ interpolation points (i.e., the interpolant is highly oscillatory).
2. The proposed spectrum-adapted weighted least squares method tends to outperform the Lanczos method for matrices such as si2 and cage9 that have a large number of distinct interior eigenvalues.

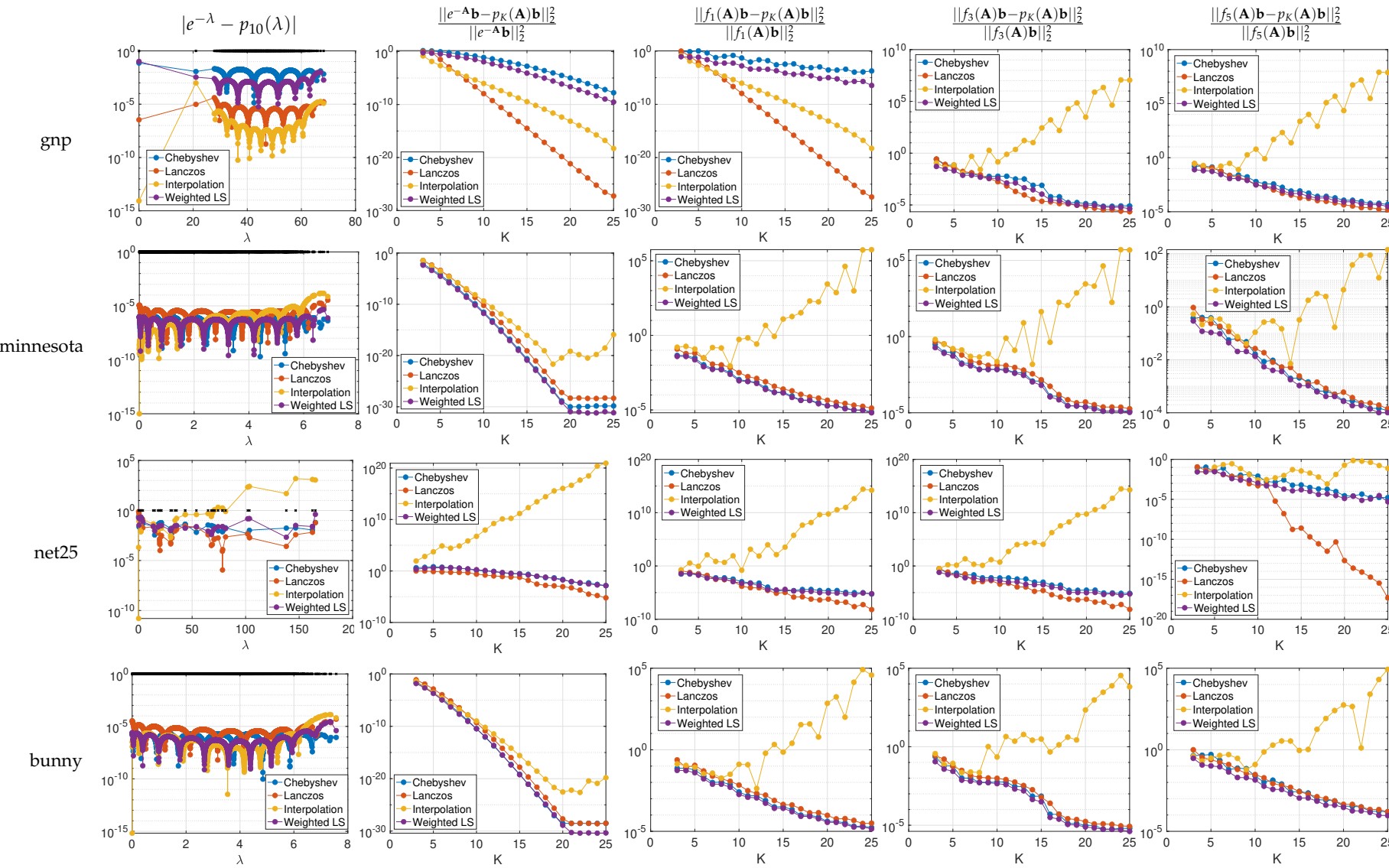

**Figure 5.** *Cont.*

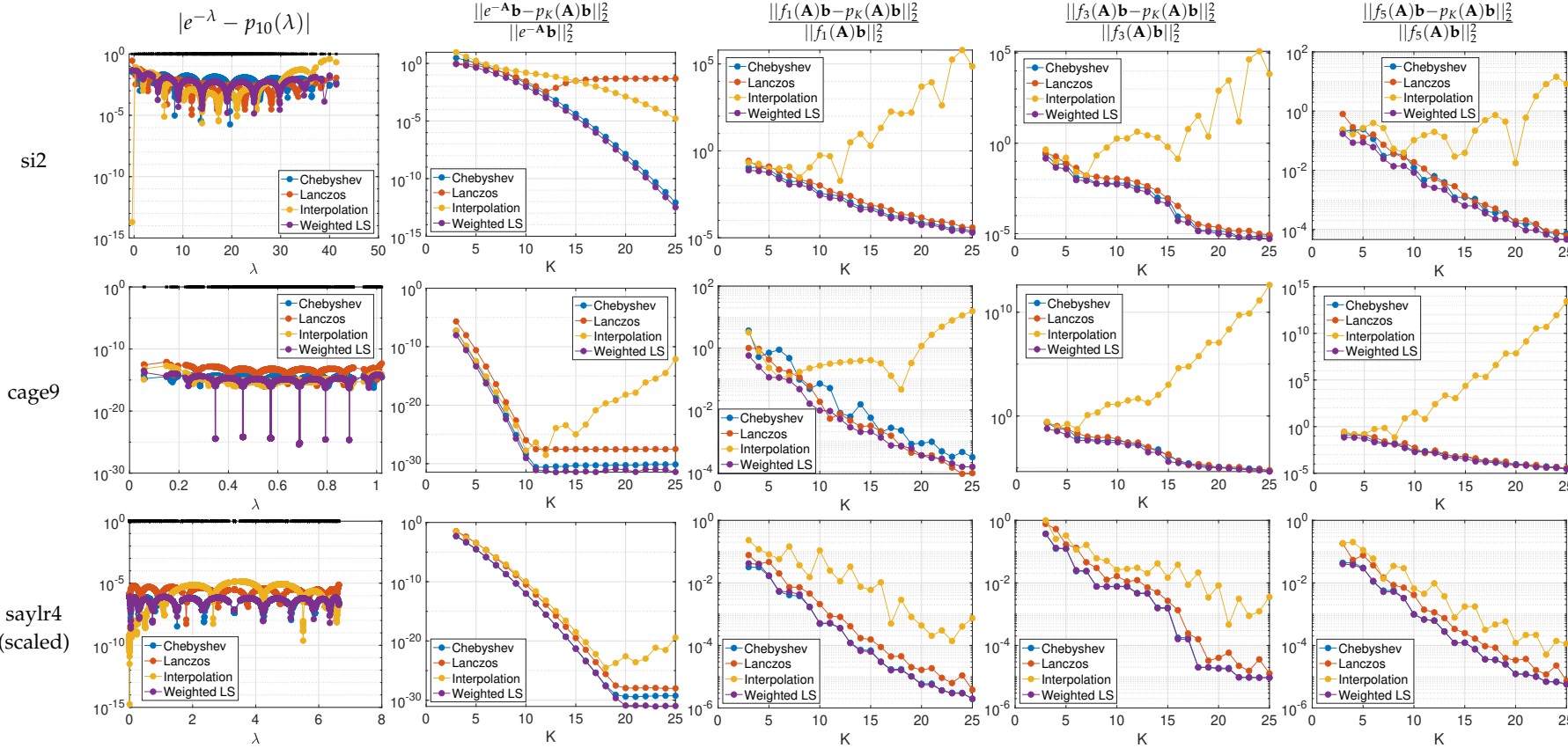

**Figure 5.** Approximations of $f(\mathbf{A})\mathbf{b}$ with $\mathbf{b} = \mathbf{V1}$ and $f(\lambda)$ equal to $e^{-\lambda}$ (first two columns) and lowpass, bandpass, and highpass spectral graph filters (last three columns, respectively).

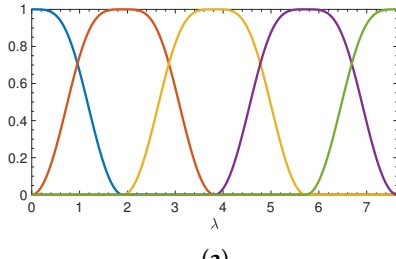

(**a**)

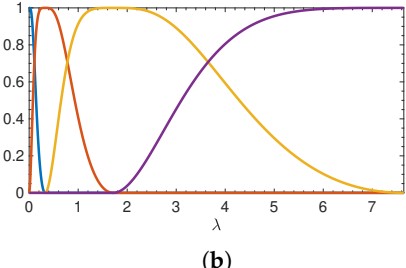

(**b**)

**Figure 6.** Spectral graph filter bank examples. (**a**) A set of five uniform translates of an itersine kernel $\sin\left(\frac{\pi}{2}\cos^2(\pi x)\right)$ [46,57]. (**b**) A set of four log-warped translates (also called octave-band or wavelet filters) [46]. We plot both sets of spectral graph filters on the spectrum of the Stanford bunny graph with $N = 2503$ vertices; however, the design only depends on the spectral range $[0, \lambda_{\max}]$, so the filters will look the same (except stretched) for all graphs considered, e.g., in Figure 5.

## 5. Application I: Estimation of the Norms of Localized Graph Spectral Filter Dictionary Atoms

A common method to extract information from data residing on a weighted, undirected graph is to represent the graph signal as a linear combination of building block graph signals called atoms, a collection of which is called a dictionary. In this section, we consider localized spectral graph filter dictionaries with the form $\mathcal{D} = \{\boldsymbol{\varphi}_{i,l}\}_{i=1,2,\ldots,N;\; l=1,2,\ldots,L}$, where each atom is defined as $\boldsymbol{\varphi}_{i,l} := f_l(\mathbf{L})\delta_i$ with $\mathbf{L}$ being the graph Laplacian and $\delta_i$ having a value of 1 at vertex $i$ and 0 elsewhere. Each atom can be interpreted as the result of localizing a spectral pattern characterized by the filter function $f_l$ to be centered at vertex $i$ in the graph. See [58] for more details about localized spectral graph filter dictionaries and their applications as transforms and regularizers in myriad signal processing and machine learning tasks.

For large, sparse graphs, the dictionary atoms are never explicitly computed; rather, their inner products with the graph signal are approximated by $\langle \mathbf{b}, \boldsymbol{\varphi}_{i,l} \rangle = \delta_i^\top f_l(\mathbf{L})\mathbf{b} \approx \delta_i^\top p_{l,K}(\mathbf{L})\mathbf{b}$, using polynomial approximation methods such as those described in Section 1 or those proposed in this work. However, in graph signal processing applications such as thresholding for denoising or compression [58–60] or non-uniform random sampling and interpolation of graph signals [41,45,58,61], it is often important to form a fast estimate of the norms of the dictionary atoms, $\{||\boldsymbol{\varphi}_{i,l}||_2\}_{i,l}$. Since $||\boldsymbol{\varphi}_{i,l}||_2^2 = \langle \boldsymbol{\varphi}_{i,l}, \boldsymbol{\varphi}_{i,l} \rangle = \delta_i^\top f_l^2(\mathbf{L})\delta_i$ is a bilinear form of the type $\mathbf{u}^\top f(\mathbf{A})\mathbf{u}$, the norm of a single atom can be estimated via quadrature methods such as Lanczos quadrature [53] (Ch. 3.1.7), [56] (Ch. 7), [62]; however, doing this for all $NL$ atoms is not computationally tractable since each requires a different combination of function and starting vector. Other alternatives include the methods discussed in [63] for estimating diagonal elements of a matrix that is not explicitly available but for which matrix-vector products are easy to evaluate, as $||\boldsymbol{\varphi}_{i,l}||_2 = \sqrt{\left[f_l^2(\mathbf{L})\right]_{i,i}}$.

In this application example, we estimate the norms of the dictionary atoms through the products of matrix functions with random vectors, as follows. Let $\mathbf{x}$ be a random vector with each component having an independent and identical standard normal distribution (in fact, we are only utilizing the property that the random components have unit variance). Then we have

$$\mathrm{var}\left(\langle \boldsymbol{\varphi}_{i,l}, \mathbf{x} \rangle\right) = \mathrm{var}\left(\sum_{n=1}^{N} x(n)\varphi_{i,l}(n)\right) = \sum_{n=1}^{N} [\varphi_{i,l}(n)]^2 \mathrm{var}(x(n)) = ||\boldsymbol{\varphi}_{i,l}||_2^2.$$

Thus, to estimate $||\boldsymbol{\varphi}_{i,l}||_2$, it suffices to estimate $\mathrm{sd}\left(\langle \boldsymbol{\varphi}_{i,l}, \mathbf{x} \rangle\right) = \mathrm{sd}\left(\delta_i^\top f_l(\mathbf{L})\mathbf{x}\right) \approx \mathrm{sd}\left(\delta_i^\top p_{l,K}(\mathbf{L})\mathbf{x}\right)$ for a degree $K$ polynomial approximation $p_{l,K}$ to $f_l$. We therefore define each atom norm estimate as the sample standard deviation

$$v_{i,l} := \mathrm{sd}\left(\left\{\delta_i^\top p_{l,K}(\mathbf{L})\mathbf{x}^{(j)}\right\}_{j=1,2,\ldots,J}\right) \approx ||\boldsymbol{\varphi}_{i,l}||_2, \tag{10}$$

where each $\mathbf{x}^{(j)}$ is a realization of the random vector $\mathbf{x}$. In the numerical experiments, we compare the estimates resulting from Chebyshev polynomial approximation to those resulting from spectrum-adapted weighted least squares polynomial approximation. As a computational aside, in the process of estimating the spectral density via KPM in (6), we have already computed $\bar{T}_k(\mathbf{L})\mathbf{x}^{(j)}$ for each $k = 0, 1, \dots, K$ and each $j$, where $\bar{T}_k$ are the Chebyshev polynomials shifted to the interval $[0, \lambda_{\max}]$. From these quantities, we can easily compute the $p_{l,K}(\mathbf{L})\mathbf{x}^{(j)}$ vectors in (10) for different values of $l$ (different filters). See [41] (Sec. III.B.1) for details.

In the top row of Figure 7, we demonstrate the estimation of the norms of the atoms of a spectral graph wavelet dictionary generated by localizing the four filters in Figure 6b to each of the vertices of the bunny graph. Figure 7a shows the exact norms of the $NL = 2503 \cdot 4 = 10012$ dictionary atoms. In Figure 7b, we plot the estimated norms of the atoms, generated via degree $K = 8$ spectrum-adapted weighted least squares polynomial approximation with $J = 50$ random vectors in (10), against the actual atom norms. The ratios of the estimated atom norms to the actual norms are shown in Figure 7c. We show the mean of the relative error $\left| \frac{\nu_{i,l}}{\|\boldsymbol{\varphi}_{i,l}\|_2} - 1 \right|$ across all of these atoms as a single point in Figure 7d, and also repeat this experiment with different values of $K$ and $J$ and different classes of approximating polynomials, as well as for different graphs in Figure 7e–f. On all three graphs and at both values of $J$, for low degrees $K$, the estimates generated from the spectrum-adapted polynomial least squares method have lower mean relative error than those based on Chebyshev polynomial approximation. While these examples are on small to medium graphs for comparison to the exact atom norms, the method scales efficiently to dictionaries designed for large, sparse graphs.

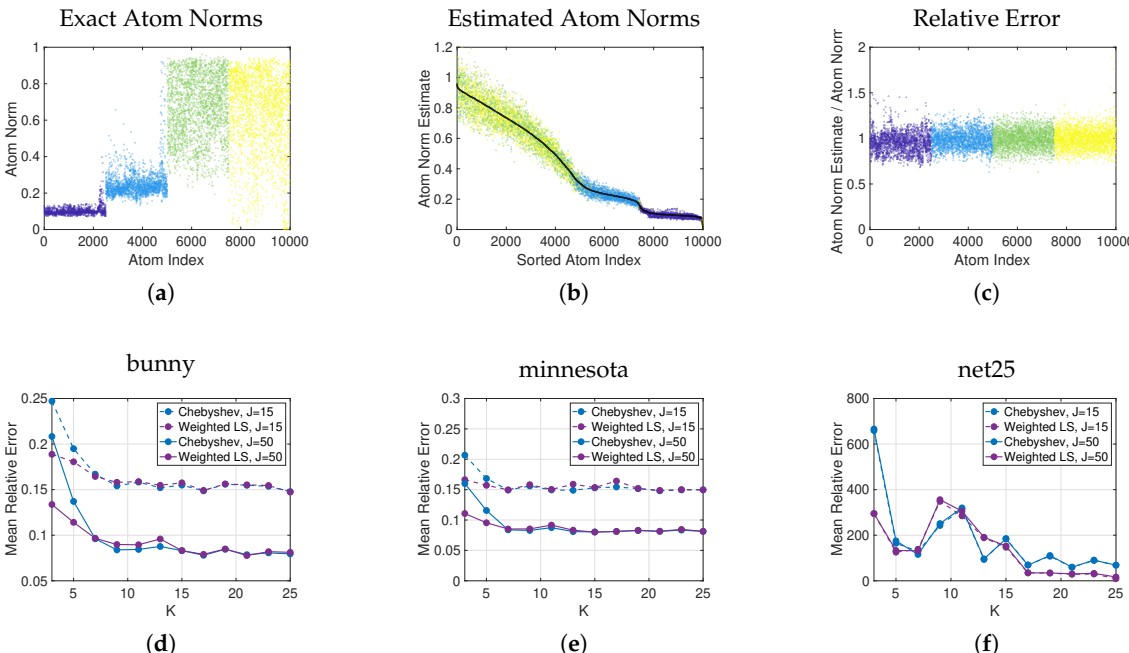

**Figure 7.** Estimation of the norms of the atoms of spectral graph wavelet dictionaries. (**a**) Exact atom norms, $\{\|\boldsymbol{\varphi}_{i,l}\|_2\}$, colored by the indices of the generating filters (shown in Figure 6), each of which is localized to every vertex on the bunny graph. (**b**) Comparison of the estimated norms, $\{\nu_{i,l}\}$, to the corresponding exact norms (shown in black). The atoms are sorted in descending order of exact norm to aid visual comparison. (**c**) The ratios of the estimated norms to the exact atom norms. (**d–f**) The mean relative errors across all atoms for dictionaries generated by the same set of filters on three different graphs, with varying polynomial approximation methods, polynomial degrees $K$, and numbers of random vectors $J$.

## 6. Application II: Fast Filtering of Time-Vertex Signals

In this section, we demonstrate the use of the spectrum-adapted approximation methods in Section 3 to accelerate the the joint filtering of time-vertex signals in both time and graph spectral domains [50,64,65].

### 6.1. Time-Vertex Signals

We consider a weighted, undirected graph $G = \{\mathcal{V}, \mathcal{E}, \mathbf{W}_G\}$ with $N$ vertices, where $\mathcal{V}$ is the set of vertices, $\mathcal{E}$ is the set of edges, and $\mathbf{W}_G$ is the weighted adjacency matrix. The combinatorial graph Laplacian is defined as $\mathbf{L}_G := \mathbf{D}_G - \mathbf{W}_G$, where $\mathbf{D}_G$ is diagonal with $\mathbf{D}_G(i,i)$ equal to the degree of the $i$th vertex. At each vertex, we observe a time series of $T$ observations. Thus, the time-varying graph signal can be represented as a matrix $\mathbf{X} \in \mathbb{R}^{N \times T}$, where $\mathbf{X}_{i,j}$ is the value on the $i$th vertex at the $j$th time. Figure 8 shows an example of a time-vertex signal.

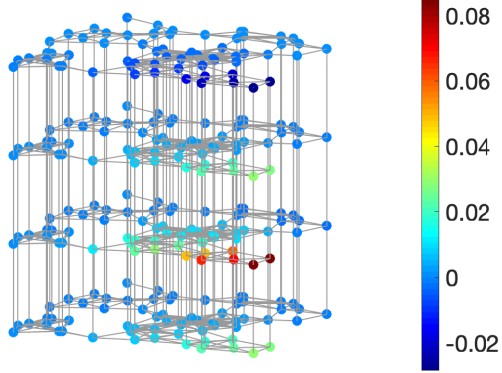

**Figure 8.** A time-vertex signal defined on a sensor graph $G$ with $N = 64$ vertices and $T = 4$ observations, visualized on a multilayer graph structure. Each layer is a copy of $G$ observed at one time point.

### 6.2. Time-Vertex Filtering

Fixing a point in time, each column of $\mathbf{X}$ is a graph signal defined on $G$. Let $\mathbf{x}_{t_j} \in \mathbb{R}^{N \times 1}$ denote the $j$th column of $\mathbf{X}$, $j = 1, \cdots, T$. Based on the graph structure of $G$, we can perform high-dimensional graph signal processing tasks on $\mathbf{x}_{t_j}$, such as filtering, denoising, inpainting, and compression [4]. In particular, for a filter $g : \sigma(\mathbf{L}_G) \to \mathbb{C}$ defined on the eigenvalues of $\mathbf{L}_G$, the graph spectral filtering of $\mathbf{x}_{t_j}$ can be computed as $g(\mathbf{L}_G)\mathbf{x}_{t_j} = \mathbf{U}_G g(\mathbf{\Lambda}_G)\mathbf{U}_G^* \mathbf{x}_{t_j}$, where $\mathbf{U}_G^*$ is the conjugate transpose of $\mathbf{U}_G$, and $\mathbf{L}_G = \mathbf{U}_G \mathbf{\Lambda}_G \mathbf{U}_G^*$ is the spectral decomposition of $\mathbf{L}_G$.

Conversely, focusing on one vertex of $G$, the $i$th row of $\mathbf{X}$ is a discrete time signal $\mathbf{x}_{v_i}^\top \in \mathbb{R}^{1 \times T}$, which indicates how the signal value changes over time on the $i$th vertex. We can compute the one-dimensional discrete Fourier transform (DFT) of $\mathbf{x}_{v_i}^\top$ by $\tilde{\mathbf{x}}_{v_i}^\top = \mathbf{x}_{v_i}^\top \overline{\mathbf{U}}_R$, where $\mathbf{U}_R$ is the normalized DFT matrix of size $T$, and $\overline{\mathbf{U}}_R$ is its complex conjugate. The DFT converts a signal from the time domain to the frequency domain, and allows for the amplification or attenuation of different frequency components of the signal. This process is referred to as frequency filtering in classical signal processing [4]. Frequency filtering of classical one-dimensional signals is equivalent to graph spectral filtering of graph signals on a ring graph [66] (Theorem 5.1). Let $\mathbf{L}_R$ denote the graph Laplacian of a ring graph with $T$ vertices. Its spectral decomposition gives $\mathbf{L}_R = \mathbf{U}_R \mathbf{\Lambda}_R \mathbf{U}_R^*$, where $\mathbf{U}_R$ comprises the DFT basis vectors (normalized to have length 1), and the $k$th eigenvalue is given by $\mathbf{\Lambda}_R(k,k) = 2 - 2\cos\frac{2\pi(k-1)}{T}$, for $k = 1, \cdots, T$.

The joint time-vertex Fourier transform of a time-varying graph signal $\mathbf{X}$ is defined as the combination of a graph Fourier transform and a DFT [50]:

$$\hat{\mathbf{X}} := \mathbf{U}_G^* \mathbf{X} \overline{\mathbf{U}}_R.$$

A joint time-vertex filter $h : \sigma(\mathbf{L}_G) \times \sigma(\mathbf{L}_R) \to \mathbb{C}$ is defined for all combinations of $(\lambda_G, \lambda_R)$ where $\lambda_G$ is an eigenvalue of $\mathbf{L}_G$ and $\lambda_R$ is an eigenvalue of $\mathbf{L}_R$. The joint time-vertex filtering is defined accordingly as

$$h(\mathbf{L}_G, \mathbf{L}_R)\mathbf{X} := \mathbf{U}_G(\mathbf{H} \circ (\mathbf{U}_G^* \mathbf{X} \overline{\mathbf{U}}_R))\mathbf{U}_R^\top, \tag{11}$$

where $\mathbf{H} \in \mathbb{R}^{N \times T}$ has entries $\mathbf{H}_{i,j} = h(\lambda_{G_i}, \lambda_{R_j})$, and $\circ$ denotes the entry-wise product of two matrices. Figure 9 shows two examples of time-vertex filters: an ideal lowpass filter

$$h(\lambda_G, \lambda_R) = \mathbb{1}_{(\lambda_G < \frac{1}{2}\lambda_{G\max}, \lambda_R < 2)}, \tag{12}$$

and a wave filter

$$h(\lambda_G, \lambda_R) = 5e^{-100\left|\arccos\left(\frac{2-\lambda_R}{2}\right) - \arccos\left(1 - \frac{\lambda_G}{2\lambda_{G\max}}\right)\right|^2}, \tag{13}$$

where $\lambda_{G\max}$ is the largest eigenvalue of $\mathbf{L}_G$. The eigenvalues of $\mathbf{L}_R$ range from 0 to 4 regardless of the size of $T$, so $\frac{1}{2}\lambda_{R\max} = 2$.

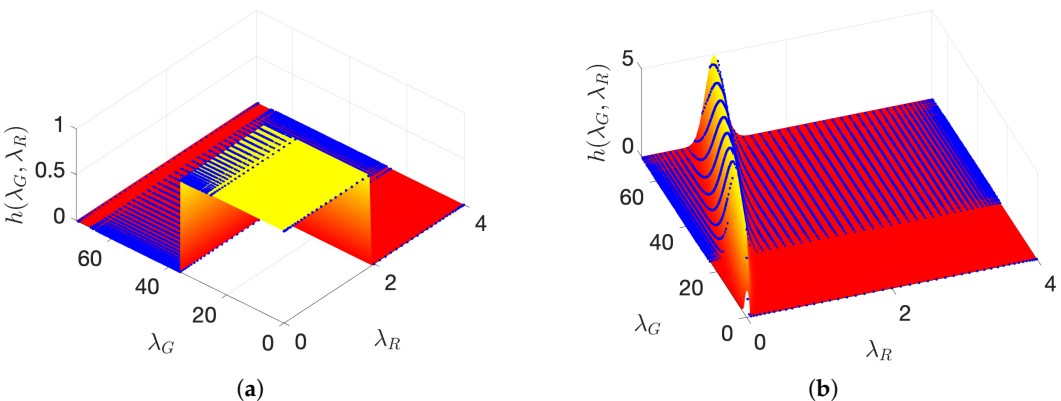

(a)                                                                      (b)

**Figure 9.** Two time-vertex filters defined for a random Erdös–Renyi graph with 500 vertices and edge probability 0.2. (**a**) An ideal lowpass filter; (**b**) a wave filter. The blue dots correspond to the positions of eigenvalue pairs.

As a two-dimensional extension of spectral filtering, the time-vertex filtering decomposes the input signal into $NT$ orthogonal components, where each component corresponds to an outer product of a graph Laplacian eigenvector and a DFT basis function. Then, the components are amplified or attenuated by the corresponding scalars $h(\lambda_G, \lambda_R)$. Finally, the scaled components are added together to obtain the filtered signal.

*6.3. Spectrum-Adapted Approximation of Time-Vertex Filtering*

Due to the high complexity of the spectral decomposition required to compute $\mathbf{U}_G$, approximation methods have been developed to accelerate joint time-vertex filtering, such as Chebyshev2D [64], ARMA2D [67], and the Fast Fourier Chebyshev algorithm [50].

As outlined in Algorithm 3, we can use the methods described in Section 3 to efficiently approximate the filtering of time-vertex signals. The overall complexity of our method is $\mathcal{O}(NT \log T + TKM)$, where $M$ is the number of nonzero entries in $\mathbf{L}_G$. The FFT of $N$ discrete-time signals of length

*T* takes $\mathcal{O}(NT \log T)$. The loop computes *T* spectrum-adapted approximations to matrix functions with polynomials of degree *K*, and thus has a complexity of $\mathcal{O}(TKM)$ for sparse $\mathbf{L}_G$.

---

**Algorithm 3** Spectrum-adapted approximate time-vertex filtering.

---

**Input** weighted undirected graph *G* with *N* vertices, time-vertex signal $\mathbf{X} \in \mathbb{R}^{N \times T}$, filter *h*
**Output** time-vertex filtered signal $\mathbf{Y} = h(\mathbf{L}_G, \mathbf{L}_R)\mathbf{X} \in \mathbb{R}^{N \times T}$

1: $\tilde{\mathbf{X}} \leftarrow$ FFT of $\mathbf{X}$, where the *n*th row of $\tilde{\mathbf{X}}$ is the 1D FFT of the *n*th row of $\mathbf{X}$, for $n = 1, \cdots, N$
2: Estimate the spectral density of *G*
3: **for** *t* in $1 : T$ **do**
4:     $f_t(\lambda_G) = h(\lambda_G, \lambda_{R_t})$
5:     Find the best order *K* polynomial approximation $p_k(\lambda_G)$ to $f_t(\lambda_G)$ via interpolation on the warped Chebyshev points (described in Section 3.1), or weighted least squares regression with evenly spaced abscissae and weights from the estimated spectral PDF (described in Section 3.2)
6:     Compute $p_k(\mathbf{L}_G)\tilde{\mathbf{x}}_t$, where $\tilde{\mathbf{x}}_t$ is the *tth* column of $\tilde{\mathbf{X}}$
7:     *tth* column of $\tilde{\mathbf{Y}} \leftarrow p_k(\mathbf{L}_G)\tilde{\mathbf{x}}_t$
8: **end for**
9: $\mathbf{Y} \leftarrow$ inverse FFT of $\tilde{\mathbf{Y}}$, where the *n*th row of $\mathbf{Y}$ is the 1D inverse FFT of the *n*th row of $\tilde{\mathbf{Y}}$
10: **return** $\mathbf{Y}$

---

*6.4. Numerical Experiments*

We consider the ideal lowpass filter (12) and the wave filter (13). We approximate $h(\mathbf{L}_G, \mathbf{L}_R)\mathbf{X}$ for both filter functions, with $T = 1000$ observations, and for different graphs *G*: gnp ($N = 500$), saylr4 ($N = 3564$) and a random sensor network ($N = 5000$), the cumulative spectral densities of which are shown in Figure 2. In each case, we choose $\mathbf{X} = \frac{1}{\sqrt{NT}}\sum_{i=1,\cdots,N;j=1,\cdots,T} \mathbf{v}_{G_i}\mathbf{v}_{T_j}^\top$, i.e., a constant vector in the joint spectral domain of $\mathbf{L}_G$ and $\mathbf{L}_R$, in order to test the average performance of the approximation methods over different combinations of eigenvalue pairs. With polynomial approximation orders ranging from $K = 1$ to 30, we follow the procedure described in Algorithm 3 to approximate $h(\mathbf{L}_G, \mathbf{L}_R)\mathbf{X}$. We estimate the cumulative spectral density functions $\tilde{P}_\lambda(z)$ with parameters $T = 10$, $J = 10$, and $K_\Theta = 30$. We use $M = 100$ in the spectrum-adapted polynomial regression/orthogonal polynomial expansion when finding the best polynomial approximation. We compare the proposed methods to the truncated Chebyshev expansion and the Lanczos method with the same polynomial order. For each method, we examine the convergence of relative errors in Frobenius norm as a function of *K* for graphs with various structures (and thus various distributions of Laplacian eigenvalues). The results are summarized in Figure 10. Similar to our observation in Figure 5, we see that the spectrum-adapted interpolation method performs well for lower polynomial orders ($K \leq 5$), but tends to be unstable at higher polynomial orders.

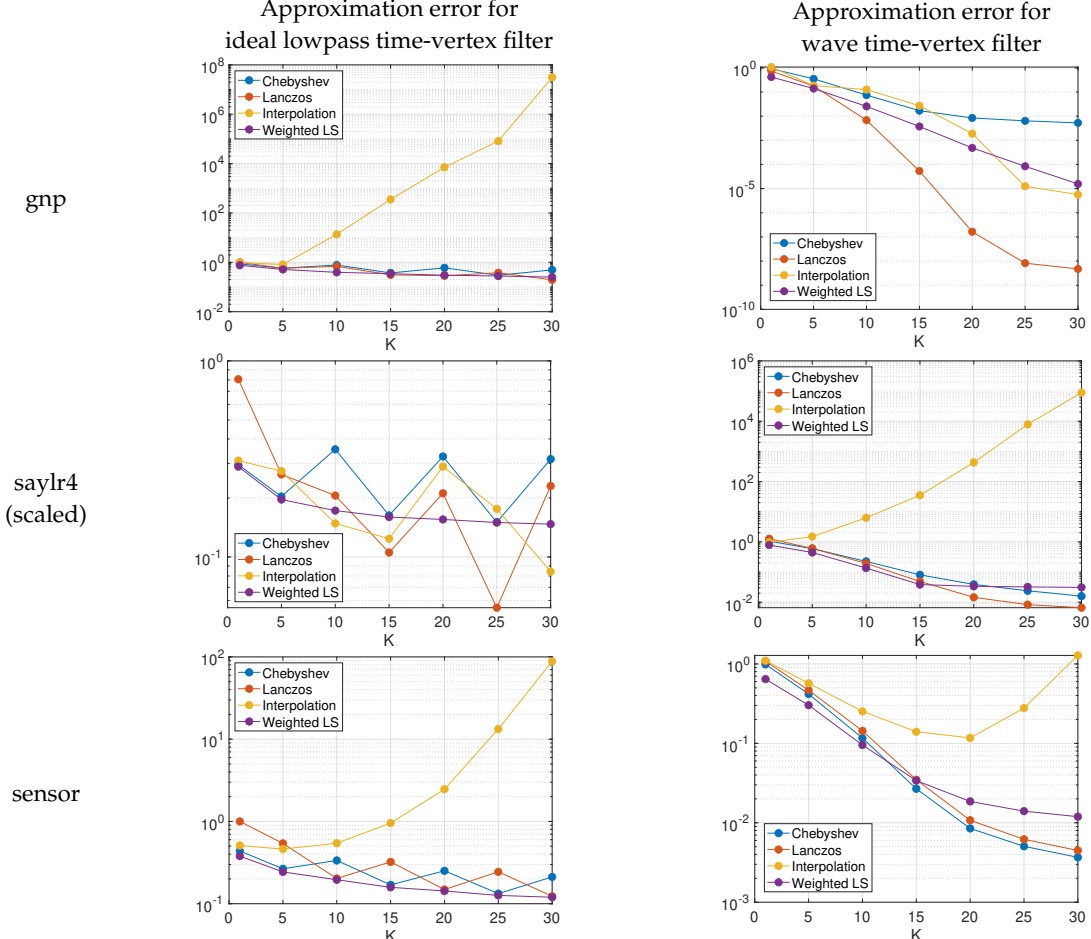

**Figure 10.** Approximation errors $\frac{||h(\mathbf{L}_G,\mathbf{L}_R)\mathbf{X}-\mathbf{Y}||_F}{||h(\mathbf{L}_G,\mathbf{L}_R)\mathbf{X}||_F}$ for $h(\mathbf{L}_G,\mathbf{L}_R)\mathbf{X}$ when $h$ is an ideal lowpass filter (12) and a wave filter (13).

*6.5. Dynamic Mesh Denoising*

Finally, we replicate a dynamic mesh denoising example presented in [50] (Sec. VI.B). The original time-varying sequence of 3D meshes of a walking dog features meshes from $T = 59$ time instances, each with $N = 2502$ points in 3D space (x-y-z coordinates). This sequence is denoted by the $2502 \times 59 \times 3$ matrix $\mathbf{X}$. The original 3D mesh from time $t = 5$ is shown in Figure 11a. The dynamic mesh sequence is corrupted by adding Gaussian noise (mean 0, standard deviation equal to $\frac{0.2||\mathbf{X}||_F}{\sqrt{2502\cdot 59\cdot 3}}$) to each mesh point coordinate. We denote the noisy 3D mesh sequence, one element of which is shown in Figure 11b, by $\mathbf{Y}$. A single graph is created by subtracting the mean $x$, $y$, and $z$ coordinates of each noisy mesh from that mesh, averaging the centered noisy mesh coordinates across all 59 time instances, and then building a 5-nearest neighbor graph on the average centered noisy mesh coordinates. The resulting graph and its spectral density function are shown in Figure 11e–f. The dynamic mesh sequence is denoised by solving the Tikhonov regularization problem [50] (Equation (30))

$$\mathbf{X}_{\text{denoised}} = \underset{\mathbf{Z}}{\text{argmin}}\, ||\mathbf{Z} - \mathbf{Y}||_F^2 + \tau_1 \text{tr}(\mathbf{Z}^\top \mathbf{L}_G \mathbf{Z}) + \tau_2 \text{tr}(\mathbf{Z}\mathbf{L}_R\mathbf{Z}^\top). \tag{14}$$

The optimization problem (14) has a closed-form solution

$$\mathbf{X}_{\text{denoised}}(:,:,i) = h_{\text{tik}}(\mathbf{L}_G, \mathbf{L}_R)\mathbf{Y}(:,:,i),\, i = 1, 2, 3; \tag{15}$$

i.e., the joint time-vertex filtering operation defined in (11) is applied to each of the noisy x, y, and z coordinates, using the same joint non-separable lowpass filter $h_{\text{tik}}$, which is defined in the joint spectral domain as [50] (Equation (31))

$$h_{\text{tik}}(\lambda_G, \lambda_R) = \frac{1}{1 + \tau_1 \lambda_G + \tau_2 \lambda_R}. \tag{16}$$

We perform a grid search to find the values of the parameters $\tau_1$ and $\tau_2$ that minimize the relative error $\frac{||\mathbf{X}_{\text{denoised}} - \mathbf{X}||_F}{||\mathbf{X}||_F}$ averaged over multiple realizations of the noise. In Figure 11g, we show the resulting filter $h_{\text{tik}}(\lambda_G, \lambda_R)$ with $\tau_1 = 7.20$ and $\tau_2 = 0.45$ on the joint spectrum of the graph shown in Figure 11e. The dashed black line in Figure 11h shows the relative error between the denoised sequence computed exactly in (15) and the original dynamic 3D mesh sequence, averaged over 20 realizations of the additive Gaussian noise. The other three curves in the same image show the average relative error when the computation in (15) is approximated by the fast Fourier Chebyshev method of [50], the Chebyshev2D method of [64], and the spectrum-adapted approximate time-vertex filtering (fast Fourier weighted least squares) of Algorithm 3, for different values of the polynomial degree $K$. All three approximations converge to the exact solution and corresponding error, but at low degrees ($K \leq 10$), the spectrum-adapted fast Fourier weighted least squares yields a better approximation. Figure 11c–d display examples of the denoised mesh at a single time ($t = 5$) resulting from two of these approximations, with $K = 6$. The difference between the two is subtle, but can be seen, e.g., by scanning the top of the dog, where the mesh points form a slightly smoother surface.

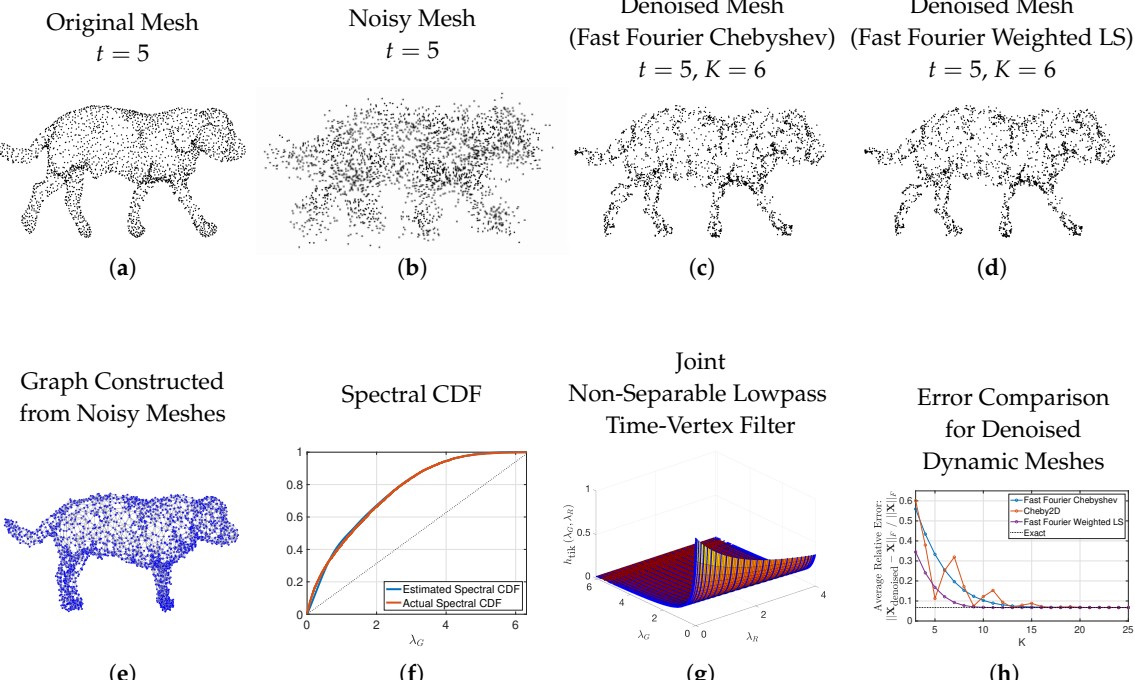

**Figure 11.** Denoising of a time-varying sequence of 3D meshes of a walking dog. Top row: one element each of the dynamic sequences of (**a**) original, (**b**) noisy, and (**c**,**d**) denoised (using two different approximation methods) meshes. Bottom row: (**e**,**f**) a 5-nearest neighbor graph (and its spectral CDF) constructed from the entire sequence of noisy meshes, which is used to denoise the meshes at all times instances; (**g**) the joint time-vertex lowpass filter (16); and (**h**) the average relative errors between the original sequence of meshes and the denoised versions attained by exactly or approximately computing (15) with different polynomial approximation methods, for a range of polynomial degrees.

## 7. Conclusions

We presented two novel spectrum-adapted polynomial methods for approximating $f(\mathbf{A})\mathbf{b}$ for large sparse matrices: spectrum-adapted interpolation and spectrum-adapted weighted least squares/orthogonal polynomial expansion. These methods leverage an estimation of the cumulative spectral density of the matrix to build polynomials of a fixed order $K$ that yield better approximations to $f$ in the higher density regions of the matrix spectrum. In terms of approximation accuracy, numerical experiments showed that, relative to the state-of-the-art Lanczos and Chebysev polynomial approximation techniques, the proposed methods often yield more accurate approximations at lower polynomial orders; however, the proposed spectrum-adapted interpolation method is not very stable at higher degrees ($K > 10$) due to overfitting. The proposed spectrum-adapted weighted least squares method performs particularly well in terms of accuracy for matrices with many distinct interior eigenvalues, whereas the Lanczos method, e.g., is often more accurate when $K$ is higher and/or the matrix $\mathbf{A}$ has many extreme eigenvalues. One potential extension would be to investigate a hybrid method that combines the Lanczos and spectrum-adapted weighted least squares approaches. We did not notice consistent trends regarding relative approximation accuracy with respect to the shape of the function $f$.

In terms of computational complexity, the cost of our methods, like Chebyshev polynomial approximation, is $\mathcal{O}(MK)$, where $M = nnz(\mathbf{A})$. For very large, sparse matrices, this complexity reduces to $\mathcal{O}(NK)$, where $\mathbf{A}$ is an $N \times N$ matrix. The Lanczos method, on the other hand, incurs an additional $\mathcal{O}(NK^2)$ cost due to the orthogonalization step, making it more expensive for large enough $K$. Finally, the proposed spectrum-adapted methods, like the Chebyshev approximation, are amenable to efficient parallel and distributed computation via communication between neighboring nodes [33]. The inner products of the Lanczos method, on the other hand, may lead to additional communication expense or severe loss of efficiency in certain distributed computation environments (e.g., GPUs).

**Author Contributions:** Conceptualization, T.F., D.I.S., S.U., and Y.S.; data curation, T.F., D.I.S., and S.U.; formal analysis, T.F., D.I.S., and S.U.; funding acquisition, D.I.S.; investigation, T.F., D.I.S., and S.U.; methodology, T.F., D.I.S., S.U., and Y.S.; project administration, D.I.S.; software, T.F., D.I.S., and S.U.; supervision, D.I.S.; validation, T.F., D.I.S., and S.U.; visualization, T.F., D.I.S., and S.U.; writing—original draft, T.F., D.I.S., and S.U.; writing—review and editing, T.F., D.I.S., S.U., and Y.S. All authors have read and agreed to the published version of the manuscript.

**Funding:** This research received no external funding.

**Acknowledgments:** We thank the anonymous reviewers for their constructive comments on an earlier version of this article.

**Conflicts of Interest:** The authors declare no conflict of interest.

**Reproducible Research:** MATLAB code for all numerical experiments in this paper is available at http://www.macalester.edu/~dshuman1/publications.html. It leverages the open access GSPBox [57].

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
