# Peer review of "Spectrum-Adapted Polynomial Approximation for Matrix Functions with Applications in Graph Signal Processing"

_algorithms, doi:10.3390/a13110295_

Round 1
Reviewer 1 Report
General comments:
This paper proposes two new methods to approximate the computation of f(A)b that has been frequently computed in signal processing and data analysis tasks. In particular, the authors propose to adapt the computation to the estimated spectrum of the matrix, in order to gain approximation accuracy in a number of scenarios. Finally, two applications on norm estimation and time-vertex filtering are provided.
The paper tackles an important computational problem and is technically sound. It is also generally well-written. The main downside is that there are several important details that are either missing or not discussed thoroughly. I would encourage the authors to consider the detailed comments below and further improve the paper.
Detailed comments (following page order):
P2 “A fourth advantage is that…”: this is a good point to make, but at this stage A is defined as a general matrix and it is little unclear how a graph might be associated with it. To make this better situated in a context, it might be better to first introduce a few common examples of A as graph-related matrices (e.g., adjacency or graph Laplacian matrix) and explain the practical meaning of f(A)b in those cases.
P3: please define the indicator function (as I understood it) in (4).
P3: it would be good to add a small note on why we would be interested in computing the inverse of approximation.
P4 Section 3.1: in order for the reader to better understand the proposed method here, the authors should provide clearer explanations on some technical details, for example: 1) what’s the intuition behind the warping being considered here? 2) how is the degree K polynomial interpolation done exactly? 3) how is p_k(A)b computed in this case (if the recursive formula on P2 is used then this should be pointed out)?
P5 first paragraph: it would be good to have some short discussions about the respective advantages/disadvantages of the four methods of setting the points {x_m} and weights {w_m}.
P4-6: the method proposed in 3.2 is not described clearly enough - is the regression problem solved explicitly (if yes, how)? How is this linked and contributing to the computation of p_k(A)b using the formula on P6 (my understanding is that if the polynomial approximation for f is found then using the recursive formula on P2 would be enough)? Furthermore, this formula itself is not immediately obvious and may benefit from further explanation.
Section 3: in general, for the reader to clearly understand the detailed steps of the two proposed methods, it would be good if the authors could provide two clear Algorithm tables (even if it’s in Appendix).
P8-9: for results in Figure 6, can the authors comment more on the ill-conditioning that makes the spectrum-adapted interpolation method unstable? Also, why do the proposed methods work better for smaller spectrum width? Finally, what’s the intuition that the spectrum-adapted weighted LS method tends to work better for graphs with large number of distinct interior eigenvalues?
P10: from Figure 7(d-f) it looks like the spectrum-adapted weighted LS method has a clear advantage for smaller K, could the authors elaborate more on this observation hence emphasise the benefit of the proposed method?
P11: do {}^\asterisk and \bar{} both represent complex conjugate? Also, in (1) and (9) the transpose {}^T is used. Given that the graph Laplacian is symmetric with real edge weights, I wonder if it would be better to make these notations more consistent in order to avoid potential confusion.
P11: I understand that the standard DFT basis corresponds to the GFT basis of a ring graph, but the temporal relationship is more intuitively captured by a path (and not a ring) graph. It would be good if the authors could add a small note on this point.
Algorithm 1: in step 1, is it a 2D FFT or an 1D FFT applied to each row of X? This should be clarified (the same goes to step 9).
P13: the point of spectrum-adapted weighted LS method being more stable than standard methods needs to be discussed in more detail - what could be the main reason behind this?
P13: the statement “subtracting the mean of each noisy mesh” is a bit unclear - from which is the mean subtracted? My understanding is that the graph is build simply on the average noisy mesh coordinates (obtained by averaging across all time instances).
P15: the difference between Figure 11(c) and (d) is not very visible, could the authors comment on that (e.g., where the lower error is reflected)?
P18: in Reference 63 the year information is missing.
Reviewer 2 Report
This manuscript is sound, well-written and the topic is interesting, so I find myself with very little to say.
• I have an impression that high-degree (weighted) polynomial interpolation/regression is probably very vulnerable to a mislocalisation of the eigenvalues, because the polynomial approximation will tend to oscillate a lot in the intervals where it's weakly constrained. That may make the method a bit dangerous to use if K is high and the estimated CDF contains spurious or mislocalised jumps. It may be worth blurring the estimated CDF to reflect uncertainty, rather than using raw estimates. I may be wrong about this.
• This is almost a matter of aesthetics, but since the authors fit a piecewise polynomial to the CDF, the weighted least-squares problem can be solved analytically for a lot of functions f, and does not need to be discretised as in sec. 3.2 (i.e. the sum can be replaced by an integral under the estimated spectral measure). Even If an analytic solution is not available for f, quadrature or even a Taylor expansion can be used.
Minor:
• Please improve the colour schemes: for instance, in fig. 1, Chebyshev is red in the first figure then blue in the second, and Least Squares goes from yellow to red. Then on fig. 6 Least Squares is purple but goes red again on fig. 7.
• More of a remark: looking at the results, I feel that a possibly way of using Lanczos more efficiently is to pull out the outlying eigenvalues (which Lanczos should be good at) than use Chebyshev approximation for the bulk.
• What the authors call Hutchinson's estimator should actually be attributed to Girard (esp. since Gaussian random variates are used), the refs. are below
@techreport{girard1987algorithme, title={Un algorithme simple et rapide pour la validation crois{\'e}e g{\'e}n{\'e}ralis{\'e}e sur des probl{\`e}mes de grande taille}, author={Girard, Didier}, publisher={Informatique et Mathématiques Appliquées de Grenoble}, year={1987} }
@article{girard1989fast, title={A fast ‘Monte-Carlo cross-validation’procedure for large least squares problems with noisy data}, author={Girard, A}, journal={Numerische Mathematik}, volume={56}, number={1}, pages={1–23}, year={1989}, publisher={Springer} }
Reviewer 3 Report
The paper proposes two low-complexity numerical methods to compute the matrix-vector product f(A)b, with b being a generic vector, A being symmetric and sparse and f being smooth. Both methods approximate f(A) by a polynomial of (low) order K and put forth an iterative computation where, at each of the K steps, a product between a vector and the sparse matrix A is required. The key novelty is on the computation of the polynomial coefficients. While the optimal schemes would require knowledge of the eigenvalues of A, most widely used schemes are agnostic to the eigenvalues and implement a worst-case approach. The paper leverages existing results in matrix algorithms to estimate first the distribution of the eigenvalues and, then, designs the filter coefficients based on the estimated distribution. While finding the distribution of the eigenvalues incurs a complexity cost relative to worst-case approaches, the savings relative to the exact computation of the eigenvalues is significant, especially for large matrices. Numerical results are provided illustrating competitive performance with Lanczos methods. Examples of the practical application of the novel methods in the context of graph signal processing (GSP) are also provided.
This is a timely, well-written, and easy-to-follow paper. Numerical examples and illustrative simulations complement the theoretical contribution. I believe that the results are of interest to the applied math, machine learning and signal processing communities and, hence, I support the publication of the manuscript.
However, several issues must be addressed before publication, including:
1. The methods put forth do not really exploit the structure of b. Their goal is to approximate matrix f(A) by a low order polynomial and, then, to exploit the recursive computation using matrix-vector products. I think this should be more clearly stated. both in the abstract and the introduction.
2. The paper could have done a better job when introducing the estimator in (5), which is critical for the rest of the manuscript. Definition of all the symbols in (5) (except for \mathbf{x} and \tilde{Theta} before (5) is recommended . The definition of \Theta before (5) is mandatory. Moreover, I think writing down the expression for \tilde{\Theta } after (5) would help to understand the approach.
3. Relation with GSP.
3.a The manuscript asserts that GSP is focused on symmetric shifts, while most GSP works do deal with undirected graphs, there are also works addressing the directed case. Please rewrite the related sentences and add a couple of references.
3.b Related to the previous point, the review of the state of the art is quite comprehensive, although I think that the following reference is also relevant for the problem addressed in this paper: https://ieeexplore.ieee.org/document/7926424
3.c Frequency filtering of classical one-dimensional signals is equivalent to graph spectral filtering of graph signals on a ring graph. Please, provide a reference for this.
3.d Section 6.2. Please provide a reference for joint vertex-time filtering joint vertex-time transforms (either add a new reference or use some of the manuscripts cited later [61-64]).
4. Notation
4.a When referring to a different section, the manuscript alternates between using “Section x” and “Sec. x”, please use only one of them (recommended: “Section x”)
4.b T is used before (5) for the number of xi_i. Later on, \mathbf{T}_K is used for a Jacobi matrix (see equation (7)). I think it would be better to use a different notation T, the number of \xi_i points. Related to this latter point, when introducing \xi_i for the first time, the authors could consider the notation $\{\xi_i\}_{i=1}^{Symbol chosen to replace T}. I believe this would help to clarify the approach.
Round 2
Reviewer 1 Report
I thank the authors for addressing my comments in detail. I am happy for the manuscript to be published.
On page 6 of the revised manuscript, the sentence “which may occur when using points…” suggests warped points obtained as described in 3.1 may not be suitable in this scenario. I presume this was not an issue for the method in 3.1 but it could be good to clarify this a bit, so that the reader understands when warped points are good to use.
Reviewer 2 Report
Ready to publish